# Deconstructing Distributions: A Pointwise Framework of Learning

**Gal Kaplun**[*]
Harvard
galkaplun@g.harvard.edu

**Nikhil Ghosh**[*]
UC Berkeley
nikhil_ghosh@berkeley.edu

**Saurabh Garg**
Carnegie Mellon University
sgarg2@andrew.cmu.edu

**Boaz Barak**
Harvard
b@boazbarak.org

**Preetum Nakkiran**
UC San Diego
preetum@nakkiran.org

## Abstract

In machine learning, we traditionally evaluate the performance of a single model, averaged over a collection of test inputs. In this work, we propose a new approach: we measure the performance of a collection of models when evaluated on a *single input point*. Specifically, we study a point's *profile*: the relationship between models' average performance on the test distribution and their pointwise performance on this individual point. We find that profiles can yield new insights into the structure of both models and data—in and out-of-distribution. For example, we empirically show that real data distributions consist of points with qualitatively different profiles. On one hand, there are "compatible" points with strong correlation between the pointwise and average performance. On the other hand, there are points with weak and even *negative* correlation: cases where improving overall model accuracy actually *hurts* performance on these inputs. As an application, we use profiles to construct a dataset we call CIFAR-10-Neg: a subset of CINIC-10 such that for standard models, accuracy on CIFAR-10-Neg is *negatively correlated* with accuracy on CIFAR-10 test. This illustrates, for the first time, an OOD dataset that completely inverts "accuracy-on-the-line" (Miller et al., 2021).

## 1 Introduction

A central question in machine learning is: what are the machines learning? ML practitioners produce models with surprisingly good performance on inputs outside of their training distribution—exhibiting new and unexpected kinds of learning such as mathematical problem solving, code generation, and unanticipated forms of robustness[1]. However, current formal performance measures are limited, and do not allow us to reason about or even fully describe these interesting settings.

When measuring human learning using an exam, we do not merely assess a single student by looking at their final grade on an exam. Instead, we also look at performance on individual questions, which can assess different skills. And we consider the student's improvement over time, to see a richer picture of their learning progress. In contrast, when measuring the performance of a learning algorithm, we typically collapse measurement of its performance to just a single number. That is, existing tools from learning theory and statistics mainly consider a single model (or a single distribution over models), and measure the *average performance* on a single test distributions (Shalev-Shwartz & Ben-David, 2014; Tsybakov, 2009; Valiant, 1984). Such a coarse measurement fails to capture rich aspects of learning. For example, there are many different functions which achieve 75% test accuracy on ImageNet, but it is crucial to understand which one of these functions we actually obtain when training real models. Some functions with 75% overall accuracy may fail catastrophically on certain subgroups of inputs (Buolamwini & Gebru, 2018; Koenecke et al., 2020;

---

[*]Equal contribution.

[1]For example, Devlin et al. (2018); Brown et al. (2020); Radford et al. (2021); Hendrycks et al. (2021a; 2020a; 2021c).

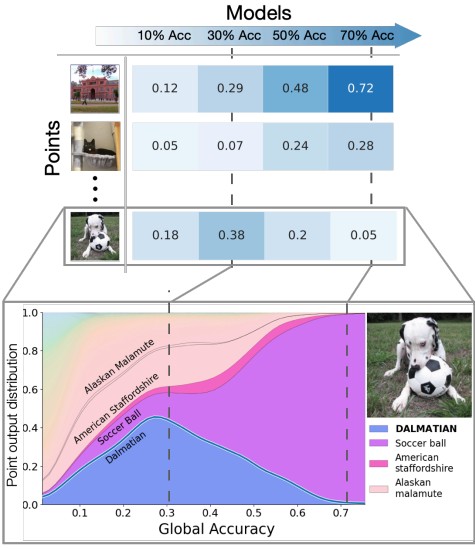

Figure 1: **Learning Profiles.** We consider the "input points vs. model" matrix of accuracies (i.e., probabilities of correct classification), with rows corresponding to inputs and columns corresponding to models from some parameterized family, sorted according to their *global accuracy*. A 70%-accurate model is on average more successful than a 30%-accurate one, but there are points on which it could do *worse*. In this case, the softmax probabilities of the bottom image show that only higher accuracy models recognize the existence of the soccer ball, throwing them off the "Dalmatian" label. Label noise or ambiguity is the reason behind some but not all such "accuracy non-monotonicities".

Hooker et al., 2019); yet other functions may fail catastrophically on "out-of-distribution" inputs. The research program of understanding models as functions, and not just via single scalars, has been developed recently (e.g. in (Nakkiran & Bansal, 2022)), and we push this program further in our work.

Figure 1 illustrates our approach. Instead of averaging performance over a distribution of inputs, we take a "distribution free" approach, and consider *pointwise* performance on one input at a time. For each input point $z$, we consider the performance of a collection of models on $z$ as a function of increasing resources (e.g., training time, training set size, model size, etc.). While more-resourced models have higher global accuracy, the *accuracy profile* for a single point $z$—i.e., the row corresponding to $z$ in the points vs. models matrix—is not always monotonically increasing. That is, models with higher overall test accuracy can perform worse on certain test points. The pointwise accuracy also sometimes increases faster (for easier points) or slower (for harder ones) than the global accuracy. We also consider the full *softmax profile* of a point $z$, represented by a stackplot on the bottom of the figure depicting the softmax probabilities induced on $z$ by this family of models. Using the softmax profile we can identify different types of points, including those that have non-monotone accuracy due to label ambiguity (as in the figure), and points with softmax entropy non-monotonicity, for which model certainty *decreases* with increased resources. And since our framework is "distribution free," it applies equally well to describe learning on both in-distribution and "out-of-distribution" inputs.

## 1.1 OUR CONTRIBUTIONS

In this paper, we initiate a systematic study of *pointwise* performance in ML (see Figure 1). We show that such pointwise analysis can be useful both as a conceptual way to reason about learning, and as a practical tool for revealing structure in learning models and datasets.

**Framework: Definition of learning profiles (Section 2.1).** We introduce a mathematical object capturing pointwise performance: the "profile" of a point $z$ with respect to a parameterized family of classifiers $\mathcal{T}$ and a test distribution $\mathcal{D}$ (see Section 2.1). Roughly speaking, a profile is the formalism of Figure 1—i.e., mapping the global accuracy of classifiers to the performance on an individual point.

**Taxonomy of points (Section 3).** Profiles allow deconstructing popular datasets such as CIFAR-10, CINIC-10, ImageNet, and ImageNet-R into points that display qualitatively distinct behavior (see Figures 3 and 4). For example, for *compatible points* the pointwise accuracy closely tracks the global accuracy, whereas for *non-monotone* points, the pointwise accuracy can be *negatively correlated* with the global accuracy. We show that a significant fraction standard datasets display noticeable non-monotonicity, awhich is fairly insensitive to the choice of architecture.

**Pretrained vs. End-to-End Methods (Section 3.2).** Our pointwise measures reveal stark differences between pre-trained and randomly initialized classifiers, even when they share not just identical architectures but also *identical global accuracy*. In particular, we see that for pre-trained classifiers

the number of points with non-monotone accuracy is much smaller and the fraction of points with non-monotone softmax entropy is vanishing small.

**Accuracy on the line and CIFAR-10-NEG (Section 4).** Using profiles, we provide a novel pointwise perspective on the accuracy-on-the-line phenomenon of (Miller et al., 2021). As an application of our framework, we construct a new "out-of-distribution" dataset CIFAR-10-NEG: a set of 1000 labeled images from CINIC-10 on which performance of standard models trained on CIFAR-10 is *negatively correlated* with CIFAR-10 accuracy. In particular, a 20% improvement in test accuracy on CIFAR-10 is accompanied by a nearly 20% drop in test accuracy on CIFAR-10-NEG. This shows for the first time a dataset with low noise which completely inverts "accuracy-on-the-line."

**Theory: Monotonicity in models of Learning (Section C).** We consider three different theoretically tractable models of learning, including Bayesian inference and a few models previously proposed in the scaling law and distribution-shift literature (Recht et al., 2019; Sharma & Kaplan, 2020; Bahri et al., 2021). For these models, we derive predictions for the monotonicity of certain pointwise performance measures. In particular, all of these models imply pointwise monotonicity behaviors that (as we show empirically) are not always seen in practice.

We demonstrate that a pointwise analysis of learning is possible and promising. However, we present only an initial study of this rich landscape. In Section 5, we discuss how our conceptual framework can guide future work in understanding in- and out-of-distribution learning, in theory and practice.

## 1.2 RELATED WORKS.

The line of work on Accuracy-on-the-Line (AoL) (Recht et al., 2019; Miller et al., 2021) studies the performance of models under distribution shift, by examining the relation (if any) between in-distribution and out-of-distribution accuracy of models. Similar to us, some works examine instance behavior in training: (Zhong et al., 2021) propose studying instance-wise performance in the NLP setting, and also take expectations over ensembles of models. Our framework is considerably more general, however, and we give new applications of this general approach. (Feldman & Zhang, 2020) and (Ilyas et al., 2022) propose tools for measuring the effect of specific training samples on the pointwise accuracy of a given example for models trained with a set amount of resources. (Swayamdipta et al., 2020) analyse individual examples in the *training set* using information based on the training dynamics. (Toneva et al., 2018) look at "forgetting events", i.e., when a training examples move from being classified correctly to incorrectly, resembling our notion of non-monotonicity.

**OOD Robustness** (Hendrycks et al., 2020b; Radford et al., 2021) show that large pretrained models are more robust to distributions shift and (Desai & Durrett, 2020) show that large pretrained models are better calibrated on OOD inputs. There is a also long line of literature on OOD detection (Hendrycks & Gimpel, 2016; Geifman & El-Yaniv, 2017; Liang et al., 2017; Lakshminarayanan et al., 2016; Jiang et al., 2018; Zhang et al., 2020), uncertainty estimation (Ovadia et al., 2019), and accuracy prediction (Deng & Zheng, 2021; Guillory et al., 2021; Garg et al., 2022) under distribution shift. Our work can be seen as an extreme version of "distribution shift", using distributions focused on a single point.

**Example difficulty** Much work was made recently to understand example difficulty for deep learning (e.g., (Jiang et al., 2020; Agarwal & Hooker, 2020; Lalor et al., 2017)). Several works study deep learning (Nakkiran et al., 2019b; Baldock et al., 2021) through the lens of example difficulty to understand certain properties (e.g., generalization or uncertainty) of deep models, while others try to modify the training distribution via either removing mislabeled examples (Pleiss et al., 2020; Northcutt et al., 2021), or controlling for hardness (Shrivastava et al., 2016; Hacohen & Weinshall, 2019). The main difference with our work is that we focus on the shape of the *curve* of example accuracy with respect to a parameterized family of models.

**Model Similarity** Several works demonstrated that the best supervised models tend to make similar predictions. (Mania et al., 2019) measure the prediction agreement between standard vision models on ImageNet and CIFAR-10 and concluding that agreement levels are much higher than they would be under the assumption of independent mistakes. (Gontijo-Lopes et al., 2021) study the effect of different training methodologies on model similarity. (Nixon et al., 2020) shows high similarity between models independently trained on different data subsets. In contrast, we focus not on comparing different types of models, but rather comparing models that span a large interval

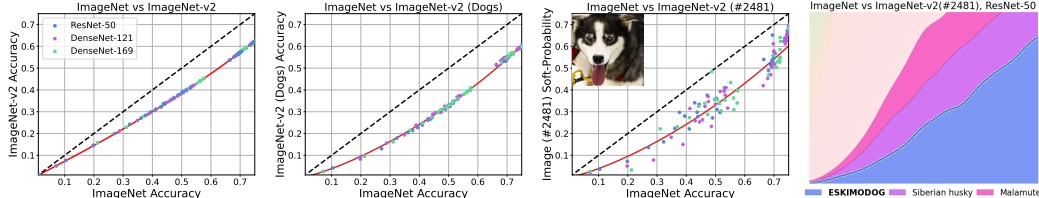

Figure 2: **Zooming In.** Average ImageNet accuracy for different models (ResNet-50, DenseNet-121, and DenseNet-169) on the x-axis with the y-axis corresponding (from left to right) to 1) ImageNet-v2 accuracy; 2) ImageNet-v2 dog-superclass accuracy; 3) the performance on a single image of a dog (i.e., the accuracy profile). The rightmost panel zooms in further and shows the *softmax-profile* of a single image for ResNet-50. This image is a "compatible point" in the sense that as global ImageNet accuracy increases, the pointwise accuracy increases, and the entropy of the softmax distribution decreases.

of accuracies. However, the above results, as well as our investigations, suggest most points' profiles remain similar under varying architectures or subsets of data.

## 2    ACCURACY-ON-THE-CURVE: ZOOMING IN

We first explore the pointwise perspective through the distribution shift from ImageNet to ImageNet-v2 (Recht et al., 2019). To start, in the left panel of Figure 2 we replicate (Miller et al., 2021) and show that for a wide variety of models, accuracy on ImageNet-v2 is well-approximated by a simple monotone function of the ImageNet accuracy. In the middle panel, we see that such a relation holds even when we consider accuracy only on the ImageNet-v2 dog super-class. That is, we *zoom-in* on the y-axis, and go from averaging over the entire ImageNet-v2 distribution to averaging over only dog classes. We see that accuracy on this sub-distribution also obeys a strong correlation with the global accuracy. This is interesting, since a priori classifiers with equally-good global performance could have very different performance on dogs.

Zooming in even further, in the third panel of this figure we evaluate the same models on an just one *individual* dog sample. That is, we compute the *accuracy profiles* (per Definition 2.2) of this particular point with respect to several parameterized learning algorithms. This example illustrates and puts into context the type of object we are interested in studying, namely general *learning profiles* which measure pointwise statistics of learning algorithms as a function of global performance.

### 2.1    FORMAL DEFINITIONS

We now formally define our central objects which are the learning profiles of a point $z = (x, y)$ with respect to some parameterized family of learning algorithms and a test distribution. These objects, visually represented in the bottom of Figure 1, capture the behavior of models from the parameterized family on $z$ as a function of their global performance on the test distribution. A *classifier* (or model) is a function $f : \mathcal{X} \to \Delta(\mathcal{Y})$ that maps an input $x \in \mathcal{X}$ into a probability distribution over the set of labels $\mathcal{Y}$. For example, for a DNN, $f(x)$ denotes the softmax probabilities on input $x$. We denote by $\hat{f}(x)$ the prediction of the classifier on $x$, obtained by outputting the highest probability label. We consider a *parameterized family* $\mathcal{T}(n)$ of algorithms, where $n$ corresponds to some measure of resources: number of samples, model size, training time, etc., and $\mathcal{T}(n)$ denotes the distribution of models obtained by running the (randomized) learning algorithm $\mathcal{T}$ with $n$ amount of resources. For the purposes of this formalism, we consider the training set to be part of the algorithm, and make no assumptions on how it is chosen or sampled. Generally, the expected performance of $\mathcal{T}(n)$ w.r.t. a global test distribution $\mathcal{D}$ will be a monotonically increasing function of $n$, and there are many works on "scaling laws" for quantifying this dependence (Rosenfeld et al., 2019; Henighan et al., 2020; Kaplan et al., 2020; Bahri et al., 2021). In particular, we consider standard training set-ups which avoid so called "double-descent" (Nakkiran et al., 2019a) pathologies when taking $n$ to be number of samples or training time. For reasons of computational efficiency, we use *training time* as our resource measure in our experimental results. However, an increasing body of works suggests that different resource measures such as time, sample size, and model size, have qualitatively similar impacts (Nakkiran et al., 2019a; 2020; Ghosh et al., 2021; Kaplan et al., 2020).

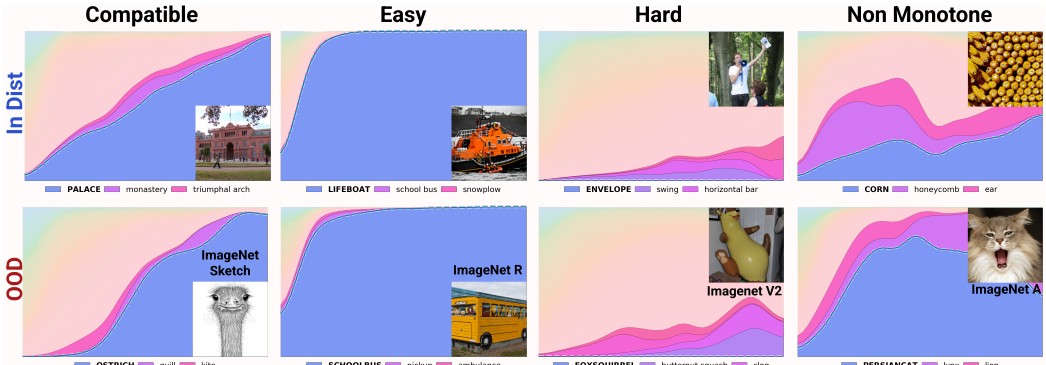

Figure 3: **Taxonomy of Samples.** Different qualitative profiles for ResNet-50 trained on ImageNet, roughly classified into 4 categories: 'Compatible' samples where sample performance traces global performance; 'Easy' samples which are classified correctly even by poor models; 'Hard' samples that even the best models fail on and; Non-monotone ones where performance behaves unpredictably w.r.t. global performance. Top row is ImageNet's validation dataset and bottom row is OOD examples. Remarkably, similar profiles emerge for both in-dist and OOD examples.

The *pointwise accuracy* $\text{Acc}_{z,\mathcal{T}}(n)$ of $\mathcal{T}$ on a point $z = (x, y)$ is the probability that the output classifier $f = \mathcal{T}(n)$ makes a correct prediction, i.e., $\hat{f}(x) = y$. The *global accuracy* $\text{Acc}_{\mathcal{D},\mathcal{T}}(n)$ of $\mathcal{T}(n)$ with respect to a distribution $\mathcal{D}$ over $\mathcal{Z} := \mathcal{X} \times \mathcal{Y}$ is the expected accuracy of points sampled from $\mathcal{D}$, i.e., $\mathbb{E}_{z \sim \mathcal{D}}[\text{Acc}_{z,\mathcal{T}}(n)]$. Throughout this paper, we will omit the test distribution $\mathcal{D}$ from subscripts when it is clear from the context. We will assume that our family is globally monotonic in the sense that $\text{Acc}_{\mathcal{D},\mathcal{T}}(n) \geq \text{Acc}_{\mathcal{D},\mathcal{T}}(n')$ for $n \geq n'$. This assumption is merely for convenience, and can be ensured e.g., by early stopping.

The *accuracy profile* of a parameterized algorithm $\mathcal{T}$ and point $z$ is the curve that maps global accuracy $p \in [0, 1]$ to the expected pointwise accuracy of $\mathcal{T}(n)$ at $z$, that is $n$ is set so the global accuracy is $p$. For example, the third panel of Figure 2 represents an accuracy profile of a particular point. Formally:

**Definition 2.1** (Accuracy profile). Let $\mathcal{T}, \mathcal{D}$ be as above. The *accuracy profile* of a point $z = (x, y)$ is the (possibly partial) function $\mathcal{A}_{z,\mathcal{T},\mathcal{D}} : [0, 1] \to [0, 1]$ that maps a global accuracy $p \in [0, 1]$ to $\text{Acc}_{z,\mathcal{T}}(n)$, where $n = n(p)$ is chosen such that $\text{Acc}_{\mathcal{T},\mathcal{D}}(n(p)) = p$.

As we will see, to get more insight on model performance we sometimes need to go beyond the accuracy and observe the full softmax probabilities induced by the model at a particular point. This motivates the following definition of *softmax profiles*, which are visually represented as stackplots in both the fourth panel of Figure 2 and bottom of Figure 1:

**Definition 2.2** (Softmax profile). Let $\mathcal{T}, \mathcal{D}$ be as above. The *softmax learning profile* of a point $z = (x, y)$ is the function $\mathcal{S}_{z,\mathcal{T},\mathcal{D}} : [0, 1] \to \Delta(\mathcal{Y})$ that maps a global accuracy $p \in [0, 1]$ to the averaged softmax distribution of predictions at $z$, among classifiers with global accuracy $p$. Specifically, with $n = n(p)$ as above, we define $\mathcal{S}_{z,\mathcal{T},\mathcal{D}}(p) := \mathbb{E}[\mathcal{T}(n)(x)]$.

We use the general name *learning profile* of a point $z$ to describe any map from $p \in [0, 1]$ to some statistics of the distribution $\mathcal{T}(n(p))(z)$. Defining learning profiles as a function of the global accuracy $p$ (as opposed to $n$), allows us to compare different resource measures on the same axis.

## 3 STRUCTURE AND DIVERSITY OF DATA AND MODELS

We now conduct a systematic study of the structure of profiles, exploring what they can teach us about data samples and training algorithms. Profiles are joint functions of an input point and a training procedure. Below, we will first fix a training procedure and vary the choice of input points: this reveals structure in data sets, through the lens of a given model. Afterwards, we will fix an input point and vary the training procedure: this reveals structure in training procedures, through the lens of a test point.

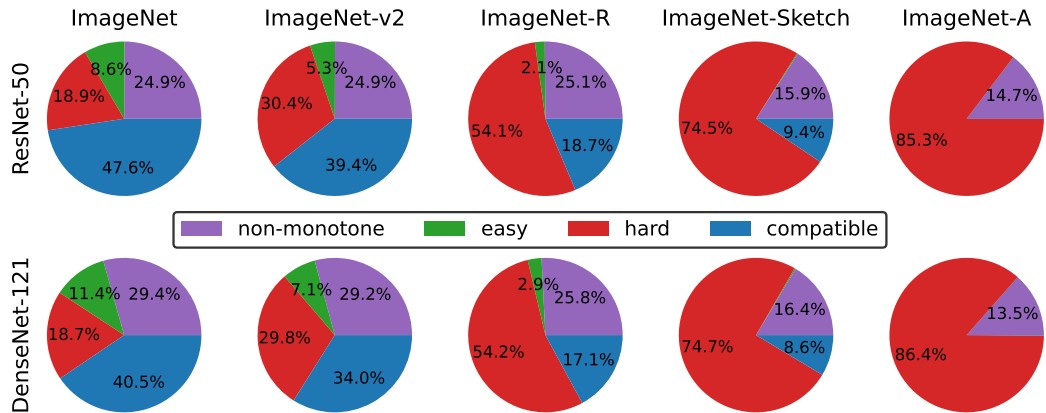

Figure 4: Each pie-chart gives a pointwise decomposition of a dataset according to profile types. Each point is classified based on the accuracy profile of an ImageNet trained classifier (see the end of Section 3.1). The decompositions are similar for both ResNet-50 and DenseNet-121 architectures.

## 3.1 STRUCTURE IN DATA

We first fix a training procedure and use the resulting profiles to study both in and out-of-distribution samples. From this analysis we broadly sketch the landscape of the various profile types. Note that the type of a point is dependent on the training procedure. Figure 3 shows several "prototypical" profiles encountered in real data and the corresponding samples. We highlight the following qualitative types:

1. **Easy points** for which even low global accuracy classifiers succeed with high probability. Note that there are out-of-distribution points which are "easy" for ResNet-50, such as the shed painted as a school-bus in Figure 3. Further, not all easy points are alike: some samples are "harder-than-average" for weak models, that become "easier-than-average" for strong models (e.g. Figure 6).

2. **Hard points** for which even high-accuracy classifiers fail. By looking at the softmax probabilities, we can disentangle the causes for the difficulty. Some points are simply ambiguous or mislabeled. For other points the softmax entropy remains high even at high global accuracies, and even the top-5 accuracy is low.

3. **Compatible points** for which the accuracy profile is close to the identity ($y = x$) function, i.e., pointwise accuracy closely tracks the average performance. It is not a priori clear that compatible points should exist. For example, one might expect the accuracy profile to always be a step function, with the individual accuracy of a sample jumping from 0 to 1 when global accuracy crosses some threshold. That is, the model could have "grokked" the sample at some global accuracy level, but performed trivially before this level (in the terminology of (Power et al., 2022)).

4. **Non-monotone points** for which the pointwise accuracy is *anti-correlated* with the global accuracy in some intervals. Again, we can use the softmax profile to better understand the potential underlying reasons for the non-monotonicity of such points. Some are mislabeled or have an ambiguous label then the classifier struggles with choosing the correct label. Other points even have non-monotone softmax entropy which implies that higher global accuracy classifiers are actually less certain about this point than lower global accuracy classifiers. The existence of non-monotone points may not always be an issue, since it is possible that powerful enough models can eventually reach high accuracy on such points. However, when individual point accuracy is important, non-monotonicity may be undesirable in resource constrained settings. Nonetheless, non-monotonicity is certainly unusual and such points may provide rich examples of interesting learning behavior.

To try to get a better quantitative understanding of dataset structure through the lens of our taxonomy on learning profiles, we use the following procedure to classify a profile as either easy, hard, compatible, or non-monotone. To evaluate if a point is non-monotone, we compute the *non-monotonicity score* of its profile, which measures the cumulative drop in pointwise performance as global performance increases (see Appendix B). If the point has a non-monotonicity score greater than 0.1, which indicates noticeable non-monotonicity, then we classify it as non-monotone. Otherwise, we classify

the profile as easy, hard, or compatible based on the $L_2$ distance of the profile to a corresponding "template" profile. Easy points are represented by the profile $f(p) = 1$, hard points by $f(p) = 0$, and compatible points by $f(p) = p$. In Figure 4, we plot the decompositions of various datasets according to the described classification of their accuracy profiles. Each profile is computed from the predictions of an ImageNet trained architecture. We see that as expected ImageNet and ImageNet-v2 contain significantly many compatible points, although there are still many points that are not most accurately described as compatible. The datasets ImageNet-R, ImageNet-Sketch, and ImageNet-A are significantly less compatible with ImageNet and are progressively harder.

The examples above are meant to illustrate the potential of the learning profiles as means of better understanding data and learning—in particular, considering entire profiles can often reveal more insight than just the final pointwise accuracy. Although the choice of profile types in our taxonomy may not be the optimal classification, we can nevertheless see that it allows us to gain insight into the structure of datasets. We hope our initial investigation can inspire future work in this area.

## 3.2 Structure in Training Procedures

Just as we can understand different samples by fixing a training procedure, we can also understand different training procedures by their behaviors on a fixed sample. Taking this viewpoint, we investigate standard architectures (ResNet-18 and DenseNet-121) on CIFAR-10, considering both models trained from scratch and those pre-trained on ImageNet. (See full experimental protocol in Appendix A). In Figure 5 we decompose the CIFAR-10 test set similarly to Figure 4, but now with the perspective of understanding model differences through the dataset. We can see that the models trained from scratch and the pre-trained models exhibit very different decompositions, but are very similar between architectures. In particular, we see that with pre-training the number of non-monotone examples decreases and points become overwhelmingly compatible. We now further probe the observed model similarity and monotonicity induced by pre-training.

**Model Similarity** We start by introducing a distance measure to compare two training procedures. Given two softmax profiles, we define the *profile distance* to be the average over all test points and accuracies $p \in [0, 1]$ of the $L_1$ distance between the softmax distributions at accuracy $p$ (see Appendix B). The rightmost panel of Figure 6 shows the pairwise profiles distances between several architectures and their pretrained variants. We find that profiles of pretrained models significantly differ from non-pretrained ones. However, controlling for the presence of pretraining, model architecture does not seem to significantly affect profiles.

**Pretraining Induces Monotonicity** We also show a specific way in which pretraining affects profiles: it drastically reduces the number of points which are *non-monotonic*. To quantify this effect, we use the *non-monotonicity score* of a profile which is the negative variation of the profile and is large when negative-slope regions exist (see Appendix B). The right panel of Figure 6 compares the CDFs of the accuracy profile non-monotonicity scores for both from scratch training and fine-tuning (see also a specific example in the middle panel). We observe that models trained from scratch are prone to significant amounts of non-monotonicity, while pretraining eliminates non-monotonicity almost completely. Further, this "elimination of non-monotonicity" by pretrained models applies for both the accuracy and entropy profiles (see Appendix Figure 10), suggesting that pretrained models display an inductive bias similar to "idealized" Bayesian inference, which always displays monotonicity (see Theorem C.1). Note that we have only considered a few specific pre-training settings and it would be interesting for future work to explore a wider range of settings to see what conclusions still hold.

## 4 Pointwise Perspective on Distribution Shifts

We now show that the pointwise perspective can shed light on distribution shifts. An important open question in this area is to understand the relationship between in- and out-of-distribution (OOD) performance, and how it depends on different factors such as pre-training. To probe this relationship, it is a common practice to evaluate methods on many OOD test sets, and measure in-dist vs. OOD performance (Recht et al., 2019; Radford et al., 2021). In several cases, these metrics are linearly correlated (after probit scaling), a phenomenon known as "accuracy-on-the-line" (Recht et al., 2019; Miller et al., 2021). However, this phenomenon does not hold universally, and we do not yet have a good understanding of when a distribution pair is linearly-correlated.

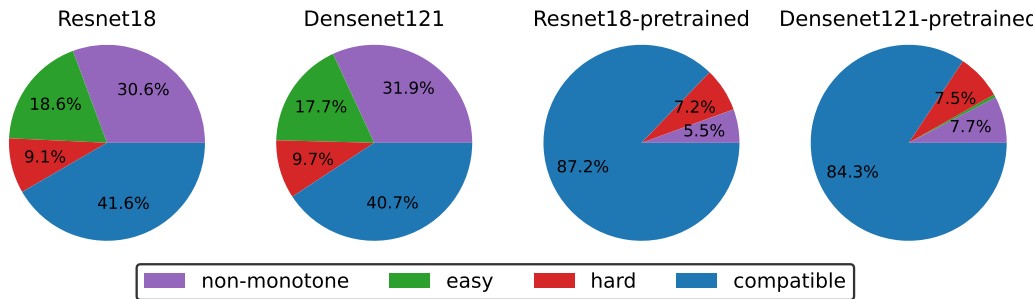

Figure 5: Taxonomy of the CIFAR-10 test set computed analogously to Figure 4 using the models trained on CIFAR-10 from Section 3.2. Pretraining significantly increases compatibility.

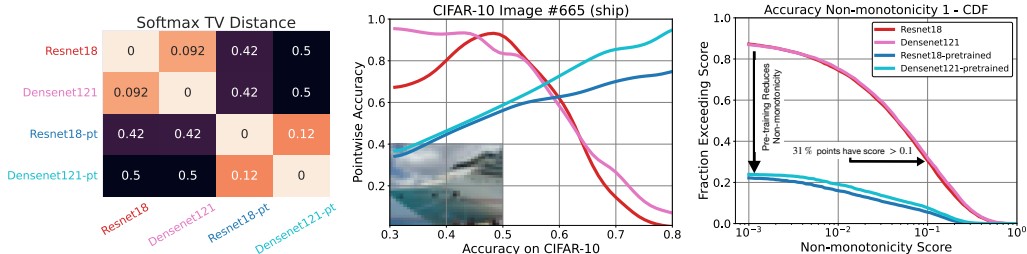

Figure 6: **Scratch vs. Pretrained.** A study of model similarity on CIFAR-10. *Left panel:* Averaged over CIFAR-10 test dataset, profiles cluster together based on pretraining and not architecture. For example, ResNet-18 profiles are much closer to DenseNet-121 ones than to ResNet-18 pretrained ones. *Middle panel:* Illustration of the effect of pretraining on a single image. *Right panel:* Pretraining reduces accuracy non-monotonicity across many points in the CIFAR-10 test set.

In the previous section, our pointwise analysis demonstrated the existence of *non-monotone* instances where pointwise and global performance are anti-correlated. Interestingly, such examples occur often enough for us to construct a non-degenerate out-of-distribution test set which breaks "the line" in much stronger ways than were previously known (see Figure 7).

Using our pointwise perspective, we construct CIFAR-10-NEG[2] (see Figure 7), a CIFAR-10-like, class balanced and correctly labeled[3] dataset of 1000 samples, which is *anti-correlated* with CIFAR-10 accuracy. Specifically, improving test accuracy by $20\%$ (from $60\%$ to $80\%$) on CIFAR-10 hurts test accuracy by $\approx 20\%$ on CIFAR-10-NEG, when training with ResNet-18 and DenseNet-121. The behavior of ResNet-18 and DenseNet-121 are largely similar as expected, and we believe the same behavior will be exhibited for standard architectures trained from scratch. In contrast, CLIP fine-tuned models on CIFAR-10-NEG are linearly correlated with CIFAR-10 performance.[4] Since the models are only trained to 80% accuracy on CIFAR-10, it is possible that the anti-correlation can be corrected at higher accuracies, however the anti-correlation exhibited is already significant as it persists for a wide range of fairly high accuracies and the drop in CIFAR-10-NEG performance is significant.

To identify a dataset of points with negative correlation, we start with the CINIC-10 test set (Darlow et al., 2018). To avoid ambiguous and mislabeled points in our new dataset, we perform CLIP-filtering: we restrict the CINIC-10 test set to points correctly predicted by a CLIP model fine-tuned on the CIFAR-10 train set. We train several ResNet-18 models on CIFAR-10 dataset and obtain per-sample monotonicity scores (defined in Appendix B) for the CINIC-10 test set. After sorting points with non-monotonicity score, we select a perfectly balanced dataset of 1000 points consisting of the top 100 most non-monotonic samples from each class. We remark that it is likely that such an approach can likely be used to construct analogous datasets which demonstrate similar anti-correlated behavior.

---

[2]Dataset: `https://anonymous.4open.science/r/CIFAR-10-NEG-F697/`

[3]Correct labeling is essential; incorrectly labeled examples will naturally be negatively correlated with global performance. As a heuristic, we use CLIP to filter such samples.

[4]Fully fined-tuned CLIP achieves 100% accuracy on CIFAR-10-NEG by design.

Figure 7: **CIFAR-10-NEG.** (a) We construct CIFAR-10-NEG, a clean dataset that has *negative* correlation with CIFAR-10 for standard models (e.g., ResNet-18 and DenseNet-121): improving test accuracy on CIFAR-10 *hurts* test accuracy on CIFAR-10-NEG. (b) Pretraining restores monotonicity: pretrained models (e.g., CLIP and ResNet-18 and DenseNet-121 pre-trained on Imagenet) have strong positive correlation with CIFAR-10-NEG. (c) A learning profile of one CIFAR-10-NEG example, showing how profiles reveal more about the evolution of predictions across learning. (d) Samples from CIFAR-10-NEG. We contrast CIFAR-10 samples with CIFAR-10-NEG samples in Appendix E.

Previous works observed weak correlation under distribution shift, but we are the first to observe *anti-correlation* between in-distribution and out-of-distribution performance for natural (non-adversarial) and correctly labeled images. Figure 7(b) shows a sample from this dataset. We juxtapose CIFAR-10 test set examples with more examples from CIFAR-10-NEG in Appendix E. It is an interesting question for future work to generate such datasets using more natural constructions and to better understand the source of the anti-correlated behavior.

## 5  DISCUSSION AND CONCLUSIONS

We conclude by discussing why we believe the pointwise perspective in general and learning profiles in particular are central to understanding both on- and off-distribution learning.

**Out-of-Distribution Inputs** When deploying ML systems, inputs are rarely drawn from exactly the same distribution as the train set. Many existing frameworks try to model this as a *distribution shift*: they assume test inputs are drawn from a distribution $\mathcal{D}'$, that is related to the train distribution $\mathcal{D}$ in some way (e.g. covariate shift (Heckman, 1977; Shimodaira, 2000), label shift (Lipton et al., 2018; Garg et al., 2020), or closeness in some divergence (Ben-David et al., 2010)). However, in practice the distribution is often not well-specified or indeed, a distribution at all. We often care about performance on *particular instances*: e.g., on correctly recognizing a pedestrian in this specific image. Furthermore, the inputs to our system may change in arbitrary and unmodeled ways (with weather, country, wildfires, etc). An instance-wise perspective is crucial in these settings.

**Lessons for Theory** We outline several concrete lessons for theory, and some speculative ones. Concretely, our experiments have identified arguably unexpected behaviors of real models and real datasets, which any potential theory of deep learning *must be consistent with*. For example, in Section 3 we found a significant number of real in-distribution samples on which DNNs are *accuracy non-monotone*: where networks with higher average accuracy (e.g., trained on more samples, or for more time) actually perform much worse. This behavior is impossible in many toy models of learning, as we prove in Appendix C. Thus, our experiments serve as guidelines for future theory work.

**Lessons for Practice** We expect that pointwise profiles are an interesting *new measurement* in many settings, which may reveal effects obscured by coarser metrics. For example, studying the softmax profile of a point can reveal not only a model's final accuracy on this point, but how its predictions *evolved* as it learnt, and potential causes of confusion along the way (e.g., texture bias or ambiguous objects). Our finding of *non-monotone* samples also suggests that current learning techniques are suboptimal in certain ways, but gives hope they can be improved. Specifically, non-monotone samples are those for which weaker models perform well (and thus, we know learning is possible), but stronger models for some reason regress. It may be possible to fix this "irrational" behavior in practice, since we know such samples are not fundamentally difficult. Indeed, we find that some techniques such as pretraining also eliminate most non-monotonicity—understanding why is an important question for future work.

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

# A    EXPERIMENTAL DETAILS

**CIFAR-10 Experiments**   In the experiments of Section 3.2, models trained from scratch on CIFAR-10 were trained from a random initialization using SGD with a Cosine Annealing learning rate schedule with initial learning rate $\eta = 0.01$, batch size 128, and weight decay $5 \times 10^{-4}$, for 30 epochs. The pre-trained models were first trained from scratch on the full ImageNet $32 \times 32$ (Chrabaszcz et al., 2017) training set using those same hyperparameters but for 100 epochs. For fine-tuning pre-trained models on CIFAR-10, the linear classification layer was initialized randomly and then trained using SGD with a learning rate of $\eta = 0.001$ and batch size of 128 for 3 epochs with no weight-decay. For all training, we used standard data augmentation (i.e., random horizontal flip, random crop of size $32 \times 32$ with padding size 4, and mean/std normalization). For each CIFAR-10 training run a single model was trained on a random subset of the training set (50,000 total samples) of size 10,000 for models trained from scratch and size 5,000 for pre-trained models. Profiles were evaluated on the CIFAR-10 test set (10,000 total samples) twice per epoch for scratch models and 10 times per epoch for pre-trained models. We computed profiles based on the evaluations of 50 independent runs. The final profiles were computed by performing Gaussian filter smoothing with $\sigma = 2$ and linear interpolation on an equally spaced grid of length 50. The architectures used were ResNet-18 and DenseNet-121.

**ImageNet Experiments**   For ImageNet, we train 10 randomly initialized seeds for three standard architectures: ResNet-50, DenseNet121 and DenseNet-169 for 90 epochs with SGD with momentum 0.9, weight decay of 0.0001 and learning rate schedule of [0.1, 0.01, 0.001] for 30 epochs each and batch size of 256 (128 for DenseNet-169). We use standard data augmentations (i.e., flip and random crop to 224x224 images). To produce softmax-profiles we use 30 equally spaced checkpoints (adding 10 checkpoints around learning-rates drops to increase plot resolution) evaluated both on and off distribution (ImageNet-A, ImageNet-R, ImageNet-sketch, ImageNet-v2 (Hendrycks et al., 2021b;d; Wang et al., 2019; Recht et al., 2019)).

## B  ADDITIONAL DETAILS

**Profile Distance**    We define the distance between the $\mathcal{P}$-profiles (e.g. accuracy-profiles, softmax-profiles, etc.) of training procedures $\mathcal{T}_1$ and $\mathcal{T}_2$ to be $\hat{\mathbb{E}} \int_0^1 d(\mathcal{P}_{Z,\mathcal{T}_1}(p), \mathcal{P}_{Z,\mathcal{T}_2}(p)) \, dp$, where $d$ is some distance measure on $\Delta(\mathcal{Y})$ and $\hat{\mathbb{E}}$ denotes averaging over points $Z$ in the CIFAR-10 test set. In the left panel of Figure 6 we take $d$ to be total variation (TV) distance, defined as $d(p, q) = \frac{1}{2} \sum_{y \in \mathcal{Y}} |p_y - q_y|$. Plots for other choices of $d$ are shown in Figure 11.

**Non-Monotonicity Score**    Given a profile $\mathcal{P}_z : [0, 1] \to [a, b]$, we can measure how much the curve $p \mapsto \mathcal{P}_z(p)$ deviates from being monotonically increasing by computing the *non-monotonicity score*,

$$\mathrm{nmono}(\mathcal{P}_z) = \int_0^1 \max\left\{0, -\frac{d}{dp}\mathcal{P}_z(p)\right\} dp.$$

Note that the non-monotonicity score is always non-negative. It is zero if and only if $\mathcal{P}_z$ is always increasing and is bounded by $b - a$. In the right panel of Figure 6, we plot the $1 - \mathrm{CDF}$ of the non-monotonicity scores of the accuracy profiles $\mathcal{A}_z$ on the CIFAR-10 test set. In Figure 10 we show the respective plots for the negative entropy profiles $p \mapsto -\mathbb{E}H(\mathcal{T}(n)(x))$ where $n = n(p)$ such that $\mathrm{Acc}_{\mathcal{T},\mathcal{D}}(n(p)) = p$ and the soft-accuracy profiles $p \mapsto [\mathcal{S}_z(p)]_y$ where for $\pi \in \Delta(\mathcal{Y})$, $\pi_y$ is the probability assigned to $y \in \mathcal{Y}$ under $\pi$.

**Pointwise Accuracy-on-the-Curve**    The plots in Figure 2 suggest the conjecture that families of algorithms $\{\mathcal{T}_i\}$ which have the same *global* accuracy curve, also have approximately similar pointwise accuracy profile: the pointwise accuracy $\mathcal{A}_{z,\mathcal{T}_i}(p)$ on $z$ is well approximated by a function $g_z(p)$ that only depends on the point $z$ (and not the algorithm $\mathcal{T}_i$). One way to test such a conjecture is to look at two algorithms $\mathcal{T}$ and $\mathcal{T}'$ and measure the average absolute difference of their pointwise accuracies at a given global accuracy (i.e., $d(p) = \mathbb{E}_z|\mathcal{A}_{z,\mathcal{T}}(p) - \mathcal{A}_{z,\mathcal{T}'}(p)|$). In Figure 12 we plot $d(p)$ when evaluating on $z$ from CIFAR-10.2 when training ResNet-18 and DenseNet-121 on CIFAR-10. We see that $d(p)$ is non-negligible, but still much smaller than we would expect if pointwise performance between $\mathcal{T}$ and $\mathcal{T}'$ was completely uncorrelated giving some weak evidence in support of the conjecture.

## C   MONOTONICITY IN MODELS OF LEARNING

We proceed to show that accuracy and softmax profiles of several models of learning obey natural *monotonicity properties*. This in contrast with our experimental results of Section 3 that demonstrate the existence of points with *non-monotone* accuracy and softmax entropy, in particular when training models from scratch (as opposed to fine-tuning). This mismatch between theory and practice can be interpreted in two (non-mutually exclusive) ways. One is that we need better models to capture realistic learning methods. The second is that non-monotonicity suggests sub-optimal behavior in practical methods, and as they improve we might expect profiles to become monotone. In particular, the fact that the accuracy and softmax entropies of Bayesian inference with a correct prior are monotone (see below), suggests that practical non-monotonicity might arise due to "mismatched priors". We start by defining the following three natural monotonicity properties with respect to a set of possible instances $\mathcal{Z}$ and a set of algorithms $ALG$:

1. **Accuracy monotonicity:** We say that a parameterized learning algorithm $\mathcal{T}$ satisfies *accuracy monotonocity* if $\forall z \in \mathcal{Z}$: $n \geq n' \implies \text{Acc}_{\mathcal{T},z}(n) \geq \text{Acc}_{\mathcal{T},z}(n')$. That is, improving global accuracy cannot hurt on any specific instance. We also consider a weaker version which we call a *pointwise scaling law*, whereby there are constants $C, \alpha > 0$ such that for all $z \in \mathcal{Z}$, $\text{Acc}_{\mathcal{T},z}(n) \geq 1 - C \cdot n^{-a}$ for all $n \geq n'$.

2. **Universality of instance difficulty:** We say that $ALG$ satisfies *universality of sample difficulty* w.r.t. $\mathcal{Z}$ if for all $z, z' \in \mathcal{Z}$ and all pairs of algorithms $\mathcal{T}, \mathcal{T}'$: $\text{Acc}_z(\mathcal{T}) \leq \text{Acc}_{z'}(\mathcal{T}) \implies \text{Acc}_z(\mathcal{T}') \leq \text{Acc}_{z'}(\mathcal{T}')$. That is, if $z$ is harder than $z'$ w.r.t. one algorithm in $ALG$, then it is harder than $z'$ w.r.t. all algorithms in $ALG$, implying an inherent "difficulty ordering" of a point.

3. **Entropy monotonicity:** We say that a parameterized learning algorithm $\mathcal{T}$ (that produces distributions over labels) satisfies *entropy monotonicity* if for every $(x, y) \in \mathcal{Z}$: $n \geq n' \implies \mathbb{E}H(\mathcal{T}(n)(x)) \leq \mathbb{E}H(\mathcal{T}(n')(x))$. That is, for $f$ drawn from $\mathcal{T}(n)$, the expected entropy of $f(x)$ is non-increasing as a function of the resource $n$.

All three properties are incomparable with one another, in the sense that there exist learning methods that satisfy any subset of these. The main result of this section is that several natural models of learning satisfy the above monotonicity properties. These include standard Bayesian inference (with correct priors) as well as certain "toy models" that were proposed in the literature to explain some puzzling global features of deep learning. The latter are highly simplified models designed to match certain *global* behaviors of DNNs such as scaling laws and accuracy on the line. While these models were designed to capture global phenomena, we show they also satisfy certain pointwise properties as well:

**Theorem C.1.**

*1. The "skills vs difficulty" model of (Recht et al., 2019) satisfies the* universality of instance difficulty *and* accuracy monotonicity *properties.*

*2. The "manifold partition" model of (Sharma & Kaplan, 2020; Bahri et al., 2021) satisfies the* pointwise scaling law *property.*

*3. Any general Bayesian inference model satisfies* accuracy monotonicity *and* entropy monotonicity. *Specific models, such as* Bayesian Gaussian Process *with a fixed kernel, also satisfy* universality of instance difficulty *with respect to a fixed training set.*

**The skill vs. difficulties model.**   Recht et al. present a highly simplified model for explaining distribution shift phenomena (Recht et al., 2019, Appendix B). In this model, each point $z$ has a "difficulty level" $d_z \in \mathbb{R}$. Each classifier $f$ has an accuracy function, $A_f : \mathbb{R} \to [0, 1]$ which is a monotonically non-increasing function mapping the difficulty (of some point $z$) the probability that the classifier is successful (on $z$). Note that $A_f$ only depends on the "skill" of $f$ so if $f$ and $g$ have the same skill the accuracy function will be the same. Now, for any two points $z, z'$, we have that for every $f$ output by some procedure $\mathcal{T}$, $A_f(d_z) \geq A_f(d_{z'})$ if and only if $d_z \leq d_{z'}$. Then, by definition, every model of this type satisfies *universal instance difficulty*. In their paper, they specifically considered a restricted version where the accuracy function of a classifier $f$ has the form $A_f(d) = \Phi(s_f - d)$ where $\Phi$ is the CDF of a standard normal and $s_f \in \mathbb{R}$ is a parameter measuring the "skill" of a model. In such a case, the global accuracy is a monotonically increasing function of the skill $s_f(n)$ (since increasing skill improves accuracy on every point and vice-versa), and hence

for every collection $\mathcal{T}(n)$, the skill will be an non-decreasing function of $n$, meaning that it satisfies *accuracy monotonicity* as well. In other words,

$$n \geq n' \implies s_f(n) \geq s_f(n') \implies \mathrm{Acc}_{\mathcal{T},z}(n) = \Phi(s_f(n) - d_z) \geq \Phi(s_f(n') - d_z) \geq \mathrm{Acc}_{\mathcal{T},z}(n').$$

**The partitioned manifold model.** This proof follows directly from the proof of (Sharma & Kaplan, 2020). For completeness, we sketch their argument here, simply observing that the existing proof continues to apply in the pointwise setting.

(Sharma & Kaplan, 2020) and (Bahri et al., 2021) propose tractable theoretical models to explain the ubiquity of *scaling laws*. In the notation of this paper, this is the observation that for many natural data distribution $\mathcal{D}$ and learning methods $\mathcal{T}$, the *global accuracy* $\mathrm{Acc}_{\mathcal{D},\mathcal{T}}(n)$ scales as $1 - C \cdot n^{-\alpha}$ for some exponent $\alpha$ that depends on the data distribution rather than particular features of the learning methods. Specifically, (Sharma & Kaplan, 2020) present a simple toy model, in which the concept learned is some Lipschitz function $f : [0,1]^d \to \mathbb{R}$, and they assume that $\mathcal{T}(n)$ corresponds to a piecewise linear approximation on $n = C^d$ cubes of side length $1/C$. They prove that such models satisfy a *global* scaling law with regression error scaling as $n^{-1/d}$. However, because of the symmetry between points in this model, their proof (as well as the proofs in (Bahri et al., 2021)) implies also the stronger notion of a *pointwise* scaling law.

**Bayesian inference model.** In a general *Bayesian inference* model where we are performing inference with respect to the true distribution, $\mathrm{Pr}$. For any fixed instance $x$, the label is a random variable $Y$ and we have a sequence of correlated random variables $Z_1, Z_2, Z_3, \ldots$. For every $n \in \mathbb{N}$, the $n^{\text{th}}$ posterior distribution $p_n$ is a random element in $\Delta(\mathcal{Y})$ obtained by sampling $z_1, \ldots, z_n$ and letting $p_n(y \mid z_1, \ldots, z_n)$ be the distribution of $Y \mid Z_1 = z_1, \ldots, Z_n = z_n$. The prediction algorithm after observing $z_1, ..., z_n$ is then arbitrarily choosing from the set $\arg\max_{y \in \mathcal{Y}} p_{n+1}$. Since different inference methods could correspond to completely different random variables, in general such models do not satisfy universal instance difficulty. However, they do satisfy *entropy monotonicity* and *accuracy monotonicity*. This is shown by the following lemma:

**Lemma C.2** (Entropy and accuracy monotonicity of Bayesian inference). *Let* $Y, Z_1, Z_2, \ldots$ *be defined as above, and consider the process of sampling* $z = (z_1, z_2, \ldots)$. *Then for every* $n$, *defining the posterior distribution* $p_n$ *as above,* $\mathbb{E}H(p_n) \geq \mathbb{E}H(p_{n+1})$ *and* $\mathbb{E}\|p_n\|_\infty \leq \mathbb{E}\|p_{n+1}\|_\infty$, *where the expectations are over the sampling of* $z$ *and for* $q \in \Delta(\mathcal{Y})$, $\|q\|_\infty = \max_{y \in \mathcal{Y}} q(y)$.

Lemma C.2 clearly implies the Bayesian inference satisfies entropy monotonicity. The reason it also implies accuracy monotonicity is the following: If $\|p_n\|_\infty = \alpha$ and there are $k$ labels $y_1, \ldots, y_k$ for which $p_n(y_i) = \alpha$, then given this posterior $p_n$, WLOG we will predict that the label is $y_i$ with probability $1/k$. But since we are in the Bayesian setting, we assume that the posterior correctly models the world, that is for each $i \in \{1, \ldots, k\}$ the probability the true label was in fact $y_i$ is $\alpha$, and hence the probability for correct prediction is $\sum_{i \in [k]} \mathrm{Pr}[\hat{Y} = y_i \text{ and } Y = y_i] = \sum_{i=1}^{k} \frac{1}{k}\alpha = \alpha = \|p_n\|_\infty$.

*Proof of Lemma C.2.* When $p_{n+1}$ is obtained by conditioning $p_n$ on the value $z$ of $Z_{n+1}$, then we can write $p_n = \sum_{z \in \mathrm{Supp}(Z_{n+1})} \alpha_z p_z$ where $p_z$ is obtained by conditioning $p_n$ on $Z_{n+1} = z$ and $\alpha_z = \mathrm{Pr}[Z_{n+1} = z | Z_1 = z_1, ..., Z_n = z_n]$. Hence $p_{n+1} = p_z$ with probability $\alpha_z$. But now the result follows from the concavity of entropy and convexity of the infinity norm: $\mathbb{E}H(p_n) \geq \mathbb{E}\sum \alpha_z H(p_z) = \mathbb{E}H(p_{n+1})$ and $\mathbb{E}\|p_n\|_\infty \leq \mathbb{E}\sum \alpha_z \|p_z\|_\infty = \mathbb{E}\|p_{n+1}\|_\infty$ $\qquad \square$

# D EXTRA FIGURES

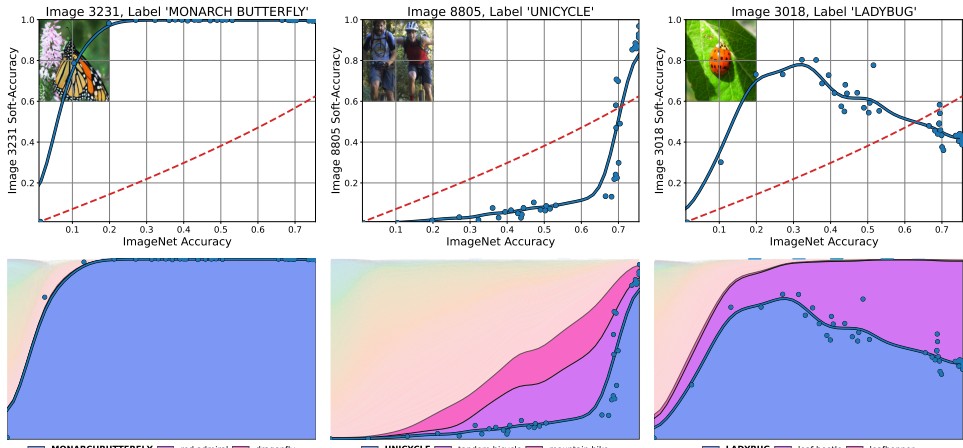

Figure 8: Example plots of softmax profiles obtained from ResNet-50 training on ImageNet, displaying a variety of interesting behaviors.

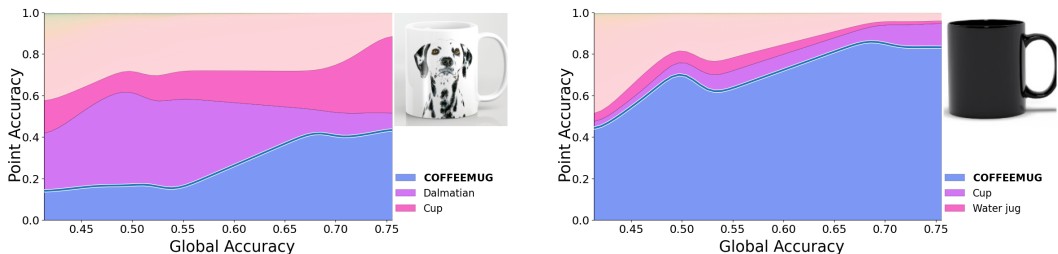

Figure 9: **Global shape vs. local features.** Lower accuracy models tend to be more sensitive to local features than the global shape of an image. For example, in this coffee mug, lower accuracy classifiers are thrown off by the illustration of a Dalmatian dog. Similar results can be seen with other images whose local texture or features conflicts with the global shape. In this sense, higher-accuracy classifiers behave more closely to humans, for whom global structure dominates local one (Navon, 1977).

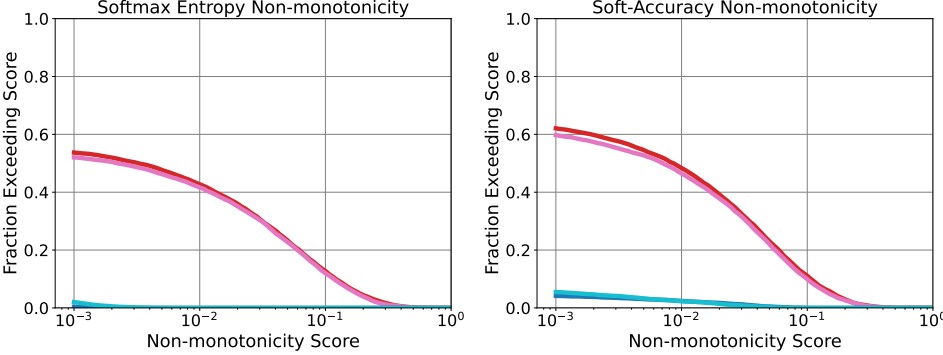

Figure 10: Plot of CDFs of the non-monotonicity score similar to the right panel of Figure 6 for the softmax entropy (*left panel*) and the soft-accuracy (*right panel*). For both measures, non-monotonicity is sharply reduced by pre-training, just as for accuracy.

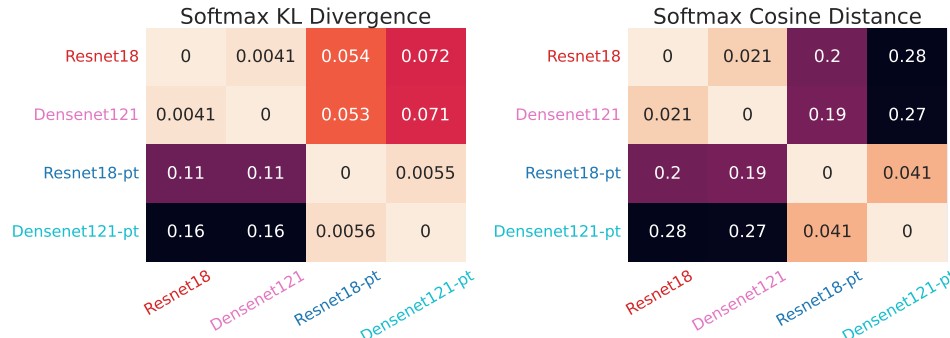

Figure 11: Heatmaps of profile distances similar to the left panel of 6 for the KL-Divergence (*left panel*) and the Cosine Distance (*right panel*). For both distances from-scratch models display higher similarity to each other than to pre-trained models.

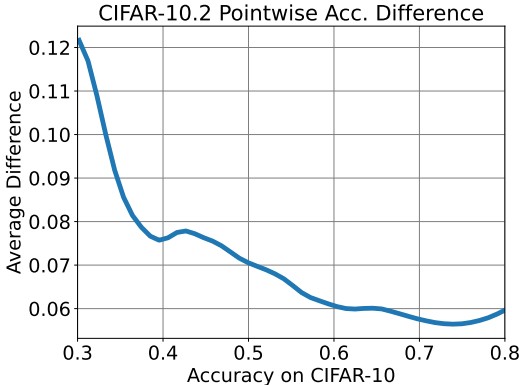

Figure 12: For each accuracy $p$ on the x-axis we plot the average pointwise absolute difference of the accuracies of a Resnet-18 model and a Densenet-121 model on the CIFAR-10.2 dataset. The values are fairly small, especially for higher accuracies.

## E    SAMPLES FROM CIFAR-10 AND CIFAR-10-NEG

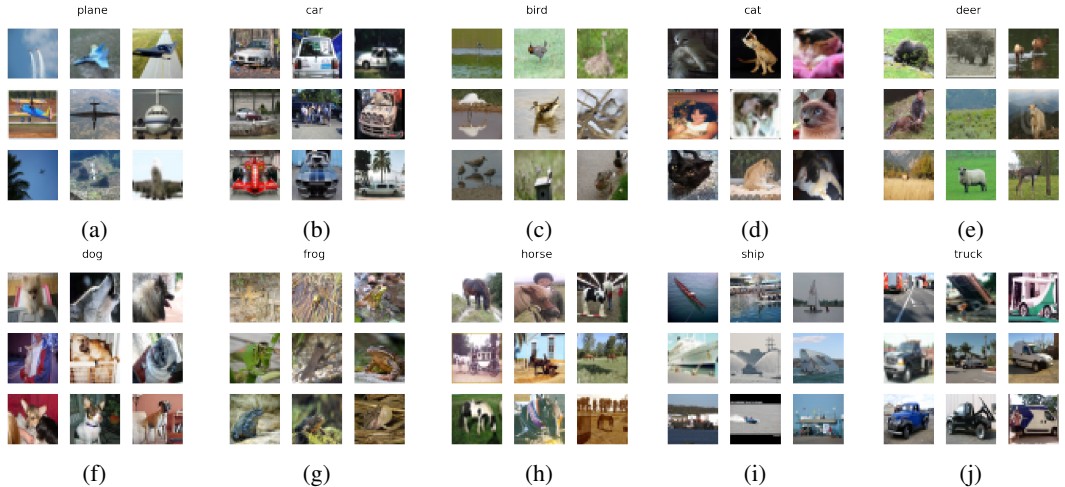

Figure 13: Random samples from CIFAR-10-NEG set.

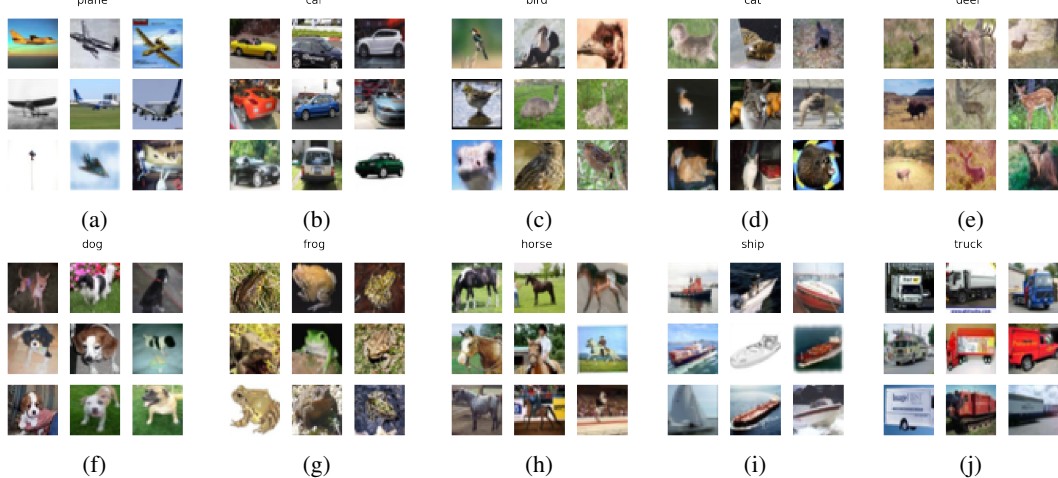

Figure 14: Random samples from CIFAR-10 test set.

