# OpenReview forum: "Deconstructing Distributions: A Pointwise Framework of Learning"
_ICLR.cc/2023/Conference — ICLR 2023 poster_

### Official Review · Reviewer_NDc7 · 2022-10-16

**Confidence:** 4
**Correctness:** 3
**Technical Novelty And Significance:** 3
**Empirical Novelty And Significance:** 4
**Recommendation:** 8

**Clarity, Quality, Novelty And Reproducibility:**

 I think the paper is very well-written and easy to understand (except for a few issues in Appendix C raised above). As far as I know, the results in the paper are novel.

**Strength And Weaknesses:**

**Edit 10-29-2022 after reading other reviews.** I think reviewer diDC brings up a really good point that I think the authors should really address -- what about the behavior observed in this paper is dissimilar from more basic situations? E.g. take the fitting of the MLE of the mean of a Gaussian with gradient descent (I know you can just take the mean and skip gradient descent, but the point is to get a series of better and better "models"). If you randomly initialize at one of the datapoints, it a reasonable fraction of datapoints will be negatively correlated with overall "model" performance (basically, everything that lies further away from the mean than the initialization). Is there more complex behavior going on that this toy example can't capture? If so, do the authors know if this more complex behavior will / will not show up in basic regression models (linear / logistic regression)?


-------------------

This paper is a little different from many machine learning papers in that it does not prove any new theory, it does not provide state of the art empirical results, and it does not answer any open questions in the field. What it does provide is some new interesting / mysterious empirical observations about the behavior of machine learning models that (to my knowledge) were not previously well known; I especially appreciated the fact that the authors have found some new ways in which toy theoretical models for explaining machine learning do not match actual machine learning models. From that perspective, I think this is a solid paper that could lead to thought provoking discussions at the conference.

I had a few scattered issues with the current draft of the paper:

- The paper only constructs families of models with increasing training accuracy by using more training time. The authors argue that other ways of constructing families of models (e.g. more complex models / more training data) might be expected to perform similarly. But there aren't experiments in the paper to demonstrate this, and I think this limitation should be emphasized more in the paper.
- I think the definitions of easy / hard / compatible / non-monotone points should go before the discussion of those different types of points. I also felt the enumerated discussion of these points was a little unnecessary, and could be cut to move more important things into the main text.
- Model similarity uses $L_1$ distance, whereas the definition of easy / hard / compatible uses $L_2$. Why the difference?
- Right under Fig 7, there is a reference to Figure 7(b). Should this be 7(c)?
- Why was 0.1 chosen as a threshold for nmono$(P_z)$?

A few comments on Appendix C
- I thought the results in here were pretty interesting. It might be worth adding a paragraph or two in the main text to summarize these results  (currently references to them are a little scattered and unclear).
- I don't think $H$ was ever defined as the entropy.
- "... obtained by sampling $z_1, \dots, z_n$ and letting $p_n(y \mid z_1, \dots, z_n)...$ I think you mean to condition on $x$ as well here?
- Is Lemma C.2 meant to hold for all $x$? This should be stated explicitly.
- "we assume that the posterior correctly models the world, that is..." This is not always true (I would say it is almost never true). It could be true asymptotically as $n \to \infty$ if the model's likelihood is correctly specified. I think this could use some more discussion.
- "Hence $p_{n+1} = p_z$ with probability $\alpha_z$." I don't think there's really a sense in which a distribution equals another distribution with a certain probability; the distributions are equal or they are not. This sentence doesn't seem necessary for the rest of the proof in any case.

**Summary Of The Paper:**

This is an empirical paper that proposes a new way to examine the predictions of machine learning models. In particular, it proposes to compare how well a variety of models predict a single, fixed test point. The authors note that models have a few surprising features with respect to this measure: in commonly used datasets, a large number of datapoints can have prediction quality negatively correlated with overall accuracy on the model's training set, and that pretraining often removes this negative correlation. The authors also note that various simplified theoretical models of machine learning training do not capture this behavior.

**Summary Of The Review:**

Overall, I think this paper demonstrates a few interesting empirical results, and would be a good addition to the community.

---

> ### Author Response · Authors · 2022-11-11
> **Author Respone**
>
> We would like to thank the reviewer for the thoughtful review and for championing the paper. We address specific comments below:
>
> - The reviewer brings about an interesting study case: When estimating the mean of a gaussian from observations using gradient descent. As GD will converge, the performance on points closer to the mean will get better while deteriorating on points farther from the mean. This toy model exhibits non-monotonicity but does not fully capture its subtleties.
>     - For example, in the first figure (see also Figure 8 right), the model is first improving on the image of the Dalmatian and then worsening on the same sample. This means, that the features the model relies on change (perhaps from local to global, see figure 9). This behavior is delicate and requires a more complex toy model to explain: one where doing things correctly (similar to pretraining) results in monotonicity and doing things sub-optimally (random initialization) results in non-monotonicity.
>     - Secondly, it does not capture entropy non-monotonicity. As GD converges and global accuracy improves the entropy (or uncertainty) about predictions will decrease. This does not happen in practice (Fig 3. right, Fig 6 middle, Fig 7 (c), Fig 8 right).
> - The question of whether and when non-monotonicity happens in linear/logistic regression is to our knowledge an open question and we are not sure about it either. We do note that our proof regarding “the partitioned manifold model” can be interpreted as 1 Nearest Neighbor being pointwise monotone up to tiny perturbations, that is, the pointwise loss can increase but in a bounded way. Particularly, when the pointwise-accuracy is high (i.e., one of the draws is not too far from the test point) the non-monotonicity should be negligible.
> - Why 0.1 was chosen? In a sense, this choice was somewhat qualitative and different options we tried yielded similar results. Our choice was empirical: substantially lower threshold would classify points that are humanly speaking monotone as non-monotone and much higher threshold would classify clearly non-monotone points as monotone.
> - We agree that the posterior correctly modeling the distribution is rarely the case. However, the point we wanted to drive is that this is not the behavior of models trained from scratch but it is of pretrained models. This bring about the curious question of what happens in pretraining that induces such a “nice” inductive bias.
> - p_{n+1} = p_z w.p. \alpha_z. Think of it as a markov chain with probability \alpha_z of returning to the same state and probability 1-\alpha_z of going to a different state.
>
> We will amend the corrections suggested by the reviewer. Some specific changes that we will make are:
>
> - More complex models or more data: we would add further discussion about this limitation.
> - L1 vs L2, good catch. It was a typo, the results are based on L1 distance.
> - 7(b) → should be 7(c).
> - We will add a paragraph in the body summarizing appendix C, we did not have one due to page limitations.
> - We will define H as entropy.
> - Lemma C.2, you can think of this result as conditioned on x. We will add a comment about it.

---

### Official Review · Reviewer_diDC · 2022-10-19

**Confidence:** 3
**Correctness:** 3
**Technical Novelty And Significance:** 3
**Empirical Novelty And Significance:** 3
**Recommendation:** 6

**Clarity, Quality, Novelty And Reproducibility:**

The clarity is overall high, quality is high, novelty is uncertain, reproducibility should be high.


**Strength And Weaknesses:**

Perhaps the greatest strength of this paper is the comprehensiveness of its investigation in terms of considering a good selection of the relevant literature and presenting a decent slice of empirical examples and theoretical considerations.  At face value the system presented for characterising the outlying data points and/or understanding the model families considered seemed novel and interesting to me; however, on further considering I found myself wondering whether the opposite was true.  I give the following criticisms from the perspective of a statistician who works with large datasets and complex models, not as a practitioner of the particular deep learning models for (e.g.) image classification to which this manuscript is addressed.

That the 'fitting' (generally considered) of certain points in a dataset degrades as the overall performance improves is in the most general sense a trivial observation.  Even in a well-specified model there will be expected that some data points are from the tails of the distribution and as we learn to represent the distribution overall these tail points naturally become 'less expected' under the overall distribution.   More relevant is the miss-specified cases where we might think of these as contaminating data points for instance, e.g. long tailed or shot noise not expected under a white noise assumption; in the literature on 'robustness' and 'robust regression' we have many ways to classify these so-called 'leverage' points and quantify their influence on the model fit (e.g. Rousseeuw & Hubert, 2011).  Similarly, in the Bayesian literature we have many visualisations and posterior predictive checking methods to do the same and stimulate model development (e.g. Gabry et al., 2017).

Admittedly, in these examples above, we are primarily considering these points with regard to the best-performing model (e.g. in a Bayesian sense, at the empirical Bayes hyper-parameter estimate), rather than in terms of a sequence including a long tail of poorly performing models.  To this end I suppose the crux of assessing this paper must come down to convincing the reviewer/reader that there is additional value in constructing these curves of training behaviour as a function of global model accuracy rather than just looking at which points are hard to fit for the best trained models (ie., best overall accuracy).  Here I would be interested in the authors' reply and the other reviewers' thoughts.


**Summary Of The Paper:**

In their manuscript entitled, "Deconstructing distributions: a pointwise framework of learning", the authors present an approach to identifying a certain class of outliers in training sets: those points for which improvements to the global model accuracy correspond to degradation in individual accuracy.


**Summary Of The Review:**

An extensive and capable investigation, which might be highly interesting although I reserve judgement for now!

---

> ### Author Response · Authors · 2022-11-11
> **Author Response**
>
> We thank the reviewer for their review, and for appreciating the strengths of our work.  We address specific comments raised by the reviewer below:
>
> Regarding curves vs. point comparisons: The reviewer is absolutely correct that the crux of our work is arguing that the entire learning profile (“curve of training behavior as a function of global accuracy”) is in fact a meaningful and important object to study, more so than just looking at individual points of the “best trained models.”
> We believe the entire curve is important for the same reason that studying asymptotics and scaling is important in many other fields, from algorithm design to statistics. Specifically, learning algorithms (and algorithms more broadly) should not be considered as methods which input a single train set, and output a single model. Rather, they should be considered as methods which can scale with the amount of input “resources” (measured in runtime, samples size, etc), and produce a sequence of increasingly-better model outputs. We then want to evaluate algorithms based on how this entire sequence of outputs scales.
> In algorithm design for example, we do not often care about the runtime of an algorithm for a fixed input size N, but rather about its runtime as we scale the input size N.
> Statistics of course takes a similar approach, where we study the asymptotic risk of methods as we scale the sample size.
> In deep learning, the study of “scaling laws” has recently been at the center of research in large language models (e.g., [2]). In all these instances, we compare methods by comparing their scaling behaviors, not just their final performance on fixed resource sizes.
>
> This is the same justification we have for studying the entire learning profile, rather than just single points (of the best-performing models): because the entire profile is the fundamental object that captures the relationship between input resources and output performance.
> The only difference in our approach, from the classical approaches, is that we emphasize measuring pointwise quantities. For example, instead of studying the asymptotic average risk (as we often do in statistics), we advocate for studying the asymptotic pointwise risk on individual test points. The latter is harder to study, but we believe important, especially in modern settings where distributions are heterogenous, and “average” behavior may obscure important pointwise behaviors.
>
> Note that in our definition of learning profile (Section 2.1), we allow for any choice of “resource” to increase: train time, sample size, model size, etc. In our experiments, for practical reasons, we have focused on “train time” as the resource, but our motivation is generic, and applies equally well to “sample size” as the resource.
>
> ----------
>
> [2] **Scaling Laws for Neural Language Models.** Jared Kaplan, Sam McCandlish, Tom Henighan, Tom B. Brown, Benjamin Chess, Rewon Child, Scott Gray, Alec Radford, Jeffrey Wu, Dario Amodei

---

### Official Review · Reviewer_5UuZ · 2022-10-25

**Confidence:** 3
**Correctness:** 4
**Technical Novelty And Significance:** 4
**Empirical Novelty And Significance:** 4
**Recommendation:** 6

**Clarity, Quality, Novelty And Reproducibility:**

This paper is novel and clearly written. The overall quality of this paper is high.

**Strength And Weaknesses:**

Strength: the paper is novel and the demonstration is pretty clear. The authors provide a lot of figures that are very helpful to understand the paper.

Weakness:

1. The explanation of why such a negative correlation phenomenon appears is not provided. I suspect that the existence of this phenomenon may depend on the specific models and dataset. A possible explanation might be that the models do not have enough many parameters and thus do not have the capability to fit all samples in the dataset. In other words, the models have to sacrifice the performance on a small portion of data to get a better overall average performance. If so, then I am afraid this phenomenon is less interesting.

2. It may be hard to identify the different types of points mentioned on page 6 beforehand since multiple models need to be trained and evaluated on each sample. This process may take a very long time, yet is not helpful to improve the average performance.

**Summary Of The Paper:**

This paper studies a new performance measuring method that measures the performance of a collection of models when evaluated on a single input point. The authors compare the models' average performance on the test distribution and their pointwise performance on an individual point. The empirical results show that for some individual test points, the average performance has a weak and even negative correlation to the performance on the individual point. Then the authors construct a dataset called CIFAR-10-NEG out of CIFAR-10 so that accuracy on CIFAR-10-NEG is negatively correlated with the accuracy on CIFAR-10 test.

**Summary Of The Review:**

I like this paper in general, due to the novelty of the topic and the new phenomenon. However, I do have some concerns about whether such a negative correlation phenomenon is common in general or only a corner case for specific models and datasets.

---

> ### Author Response · Authors · 2022-11-11
> **Author Response**
>
> We thank the reviewer for their review and for appreciating the strengths of our work.  We address specific comments raised below:
>
>  1. The explanation for the negative correlation phenomenon is an important direction of future work. We did not provide one to avoid speculation. The explanation provided by the reviewer, i.e., there are not enough parameters to fit the data, is not plausible for two reasons.
>
> - First, the trained models are quite large relative to their training datasets, reaching 100% train accuracy when training to completion. For example, ResNet-18 the architecture for the CIFAR-10 experiments has about 11 million parameters, more than enough to fit a 50k training set. The same holds as well for ResNet-50 (about 23 million parameters) and ImageNet (about 1.3 million samples).
> - Second, the pretrained models have the same architecture and capacity as the from scratch models but exhibit no non-monotonicity. So, informally speaking, a model initialized with pretrained weights will have a different “inductive bias”, where fewer non-monotone samples exist.
>
>
> 2. We agree that constructing the profiles is not an effortless endeavor. Note that our work is primarily scientific, and there is a long history in science of measurements that are expensive to make, yet scientifically valuable (even if they do not immediately improve practice). We view our work in this context: as a new kind of measurement that yields insight into machine learning.
>     Nevertheless, it is possible to improve efficiency of constructing profiles in several ways:
> - We can use saved checkpoints from old runs to do the evaluation. Since, we often save checkpoints whenever we do training we can use older for future data/experiments.
> - The profiles are consistent over different architectures. Thus, even in the case where older runs had different models, we can still average them out.
> - Even one seed is enough to have meaningful results: While we don’t have the plot in the paper, qualitatively we saw the same results without using the predictions of multiple models. The main advantage of having multiple models was to reduce variance rather than changing the conclusions of the paper.
> - If the inference is expensive over a large dataset, you could do model quantization to do the inference. Then, comparing to training costs the building of the profiles would be much cheaper (e.g., inference of a quantized model on CPU can be on par with non-quantized on GPU, see tables at the bottom of https://spell.ml/blog/pytorch-quantization-X8e7wBAAACIAHPhT).

---

### Official Review · Reviewer_JEZe · 2022-10-26

**Confidence:** 4
**Clarity, Quality, Novelty And Reproducibility:** The paper is well written and clear a…
**Correctness:** 4
**Technical Novelty And Significance:** 3
**Empirical Novelty And Significance:** 3
**Recommendation:** 8

**Strength And Weaknesses:**

The idea of studying the accuracy of fixed points with respect to changing models is interesting. There exists points which have a negative correlation with average accuracy is quite interesting. The observation about the difference between pretrained models and randomly initialized models is also quite surprising and would be interesting to look into.

However, I feel that some of the previous works have not been properly cited. [1] already had this observation that a few points are misclassified by larger models but correctly classified by smaller models and hence, this observation is not really new. The authors cite this paper but it would be good to state the differences more clearly.

[1] Are Larger Pretrained Language Models Uniformly Better? Comparing Performance at the Instance Level

**Summary Of The Paper:**

This work studies the how the accuracy of a fixed point changes when multiple models with increasing sizes are used for predictions which it defines as the learning profile of a point. They show that there can be points which are positively correlated with the average accuracy of models and there are certain points which are negatively correlated. They also observe that there are differences between pretained models and randomly initialized models. In particular, the number of points with non-monotone accuracy is much lower for pretained models  as compared to randomly initialized models.

**Summary Of The Review:**

The extensive study of point wise profiles seems interesting and the observation about the difference between pretrained and randomly initialized models is interesting.

---

> ### Author Response · Authors · 2022-11-11
> **Author Response**
>
> We thank the reviewer for championing our paper.  We list a detailed comparison with [1] below. We will expand the discussion on this paper for the final version of the paper.
>
> While [1] have similar motivation, their main focus, as noted by the reviewer is studying whether big models can perform worse than smaller ones on single points. The aim of our paper is more broad. We want to understand global distributional properties by observing pointwise scaling behavior, i.e., points’ profiles. As demonstrated by our paper there are multiple insights one can draw from the profiles that can teach us about the learning algorithm and the test data. We underscore some particular differences between the two works:
>
> 1. By conditioning on global accuracy our framework allows us to use different resources as the examining lens to individual points. For example, we can use training time (as we do in the paper) and dataset size as the measuring stick for the models being compared. As a result, our approach has multiple applications, like understanding test distributions and comparing training procedures, but also, it is computationally cheaper.
> 2. A large part of [1] deals with a fair comparison of models with different accuracies and constructing baselines that account for this discrepancy. Our framework naturally prevents this issue by said conditioning on the global accuracy. Moreover, we study both accuracy and soft-accuracy. This, for example, enabled us to observe entropy (or confidence) non-monotonicity in addition to accuracy non-monotonicity which was not observed in [1].
> 3. Finally, our work discuses the point-wise scaling performance, rather than comparing two models. This gives us a better handle of models’ behavior. Note, for instance, Figure 9. Comparing multiple models across scales, shows that lower accuracy models are more sensitive to local features while high accuracy models better attend to the global shape of an image.

---

### Author Response · Authors · 2022-11-11
**General Response**

We would like to thank the reviewers for their detailed and thoughtful feedback. We are glad to see that all the 4 reviewers recommend acceptance of our work. Especially:

1. All the reviewers appreciated the **clarity, organization, and quality of the writing/demonstration** which made the paper easy to follow (JEZe, 5UuZ, diDC, NDc7).

2. Multiple reviewers found the our **investigation's comprehensiveness** (diDC, NDc7) and the presented empirical phenomena **interesting and novel** (JEZe, 5UuZ, diDC, NDc7).


We respond to each reviewer individually in the respective threads.

---

### Decision · Program_Chairs · 2023-01-20

**Decision:**

Accept: poster

**Justification For Why Not Higher Score:**

However, there are concerns about missing explanations or insights on explaining the observed phenomena, as well as concerns about lack of discussion on how to efficiently apply this new perspective into model developments to improve prediction.

**Justification For Why Not Lower Score:**

The committee appreciates the clarity, organization, and quality of the writing/demonstration. The perspective is rather novel. The investigation conducted in the paper is comprehensive and the empirical phenomena presented are interesting and novel.

**Metareview: Summary, Strengths And Weaknesses:**

In this paper, the authors take a rather untraditional perspective and approach:  to study a fix data point's profile (assuming there are multiple models with increasing sizes are trained and used for predictions), which is the the relationship between models' average performance on the test distribution and the models' performance on this individual point. Interestingly, the authors show that there can be points which are positively correlated or negatively corrected with the average accuracy of models. Comparing between pretrained models and randomly initialized models, the authors observe in experiments that the number of points with non-monotone accuracy is much lower for pretained models as compared to randomly initialized models.

The committee appreciates the clarity, organization, and quality of the writing/demonstration. The perspective is rather novel. The investigation conducted in the paper is comprehensive and the empirical phenomena presented are interesting and novel.

However, there are concerns about missing explanations or insights on explaining the observed phenomena, as well as concerns about lack of discussion on how to efficiently apply this new perspective into model developments to improve prediction.


**Note From Pc:**

if the above contains the word "oral" or "spotlight" please see: "oral" presentation means -> notable-top-5% and "spotlight" means -> notable-top-25%. As stated in our emails, we are disassociating presentation type from AC recommendations